# YAP-independent mechanotransduction drives breast cancer progression

Joanna Y. Lee [1], Jessica K. Chang [2], Antonia A. Dominguez [3,4,5], Hong-pyo Lee [1], Sungmin Nam [1], Julie Chang[3], Sushama Varma[6], Lei S. Qi [3,5], Robert B. West[6] & Ovijit Chaudhuri [1]

Increased tissue stiffness is a driver of breast cancer progression. The transcriptional regulator YAP is considered a universal mechanotransducer, based largely on 2D culture studies. However, the role of YAP during in vivo breast cancer remains unclear. Here, we find that mechanotransduction occurs independently of YAP in breast cancer patient samples and mechanically tunable 3D cultures. Mechanistically, the lack of YAP activity in 3D culture and in vivo is associated with the absence of stress fibers and an order of magnitude decrease in nuclear cross-sectional area relative to 2D culture. This work highlights the context-dependent role of YAP in mechanotransduction, and establishes that YAP does not mediate mechanotransduction in breast cancer.

[1] Department of Mechanical Engineering, Stanford University, Stanford, CA 94305, USA. [2] Department of Genetics, Stanford University School of Medicine, Stanford, CA 94305, USA. [3] Department of Bioengineering, Stanford University, Stanford, CA 94305, USA. [4] Department of Chemical and Systems Biology, Stanford University, Stanford, CA 94305, USA. [5] Stanford ChEM-H, Stanford University, Stanford, CA 94305, USA. [6] Department of Pathology, Stanford University School of Medicine, Stanford, CA 94305, USA. Correspondence and requests for materials should be addressed to O.C. (email: chaudhuri@stanford.edu)

Enhanced mammographic density, associated with a tenfold increase in extracellular matrix (ECM) stiffness, is one of the strongest risk factors for breast cancer progression (Fig. 1a)[1–6]. Previous studies show that increased ECM stiffness promotes a proliferative and invasive phenotype in mammary epithelial cells[7–10]. During breast cancer progression, cancer cells invade through the basement membrane (BM) allowing metastatic dissemination to begin[11], resulting in decreased patient survival. Thus, there is a critical need to understand how enhanced ECM stiffness drives invasion.

Yes-associated protein (YAP), a transcriptional regulator that is deregulated in diverse cancers, has been implicated as a universal mechanotransducer[12]. Mammary epithelial cells (MECs) cultured on increasingly stiff 2D polyacrylamide (PAM) gel substrates show YAP accumulation in the nucleus, activating expression of YAP target genes (Fig. 1b, c)[13]. On stiff 2D substrates, stress fibers mediate flattening of the nucleus, which results in stretching of nuclear pores and YAP accumulation in the nucleus[14–16]. However, cell morphology and signaling are significantly altered by culture dimensionality[17–20], and 3D culture has been reported to be crucially important when modeling breast cancer[19]. In fact, several recent studies implicate YAP as a tumor suppressor during in vivo breast cancer[21–23]. As such, the role of YAP in mechanotransduction during breast cancer is unclear.

Here, we examine the role of YAP in mechanotransduction using breast cancer patient samples and mechanically tunable 3D culture models of the mammary epithelium. Strikingly, we find a lack of YAP activity with increased 3D culture and in vivo stiffness, in contrast with 2D PAM controls. Mechanistically, this discrepancy is related to the absence of stress fibers and a significant decrease in nuclear cross-sectional size between cells under 3D and 2D conditions. Together, these studies reveal that breast cancer progression is regulated by a YAP-independent mode of mechanotransduction.

## Results

**YAP-independent mechanotransduction in patient samples.** To determine if YAP is responsible for mechanotransduction during breast cancer invasion we examined DCIS patient samples, a carcinoma state marked by increased ECM stiffness preceding BM invasion. Immunohistochemical (IHC) stains of patient samples show that YAP does not localize to the nucleus in DCIS samples (Fig. 1d, e). In addition, 3SEQ analyses of patient samples for canonical YAP target genes (Fig. 1f) and additional YAP targets (Fig. 1g) show a lack of YAP activation. Further, analyses of publicly accessible gene expression datasets similarly showed no increase in YAP target gene expression with breast cancer (Supplementary Fig. 1a–e). However, expression of a subset of YAP target genes was increased in IDC samples (Supplementary Fig. 1f), which occurs post-BM invasion, suggesting that YAP activation may be relevant to post-invasion stages of breast cancer. Together, these analyses of three independent sets of patient data establish that YAP is not activated during early stages of breast cancer, when increased stiffness is reported to drive invasion.

**YAP-independent mechanotransduction in 3D culture.** We next examined whether YAP is responsive to increased ECM stiffness using a mechanically tunable 3D culture model of the mammary epithelium. Traditional mechanically tunable 3D culture models commonly incorporate col-1, which is highly relevant to post-invasion IDC[20]. However, col-1 is not present in the BM and can activate tumorigenic signaling independently of stiffness[24,25]. Therefore, to mimic increased stiffness in a BM microenvironment without confounding col-1 signaling, we generated interpenetrating networks (IPNs) of reconstituted BM (rBM) with alginate[10]. Addition of $Ca^{2+}$ cross-links the alginate network, increasing matrix stiffness without altering protein concentration, matrix architecture, or pore size[10]. Elastic moduli of hydrogels ranged from ~0.04 kPa for soft to ~2 kPa for stiff gels, covering the range of stiffness observed during breast cancer progression (Fig. 1a and Supplementary Fig. 2). We also generated traditionally used rBM and col-1 gels as controls (Supplementary Fig. 2). All hydrogels were used to encapsulate MCF10A cells, a nontransformed MEC line, in 3D culture acinar formation assays (Supplementary Fig. 2). Surprisingly, cells embedded in stiff IPNs or stiff col-1 gels, conditions that robustly promoted proliferation, invasion, and other markers of malignancy (i.e., β1 integrin and p-FAK) showed cytoplasmic YAP (Fig. 2a, b and Supplementary Figs. 3 and 4). Localization of YAP paralog TAZ mirrored YAP under all hydrogel conditions (Supplementary Fig. 5). Importantly, positive-control experiments treating cells in 3D culture with nuclear export inhibitor Leptomycin B (LepB) showed strong YAP nuclear localization, similar to that of cells seeded on stiff 2D col-coated PAM gels, demonstrating that 3D cultured MCF10A cells are competent for YAP activation (Fig. 2c, d).

We explored the possibility that YAP activation requires a higher range of stiffness. Although 1–2 kPa stiff hydrogels are sufficient to induce proliferation and invasion (Supplementary Fig. 3) and are physiologically relevant for malignant mammary tissue[4,5], we generated 20 kPa hydrogels, an order of magnitude stiffer than malignant mammary tissue. 20 kPa gels similarly failed to induce nuclear localization of YAP in MCF10A cells under 3D culture conditions, in contrast to identical hydrogels and cells cultured under 2D conditions (Supplementary Fig. 6a–c). In addition, increasingly malignant MEC lines, MCF10AT, and MCF10CA1a[26], and preformed acinar structures transplanted into stiff hydrogels also did not display increased YAP nuclear localization (Supplementary Fig. 6d, e).

In addition to examining YAP localization, we assessed YAP activity through analysis of YAP phosphorylation and gene expression. Western blot (WB) analysis of cells harvested from soft and stiff IPNs showed similar levels of YAP S127 phosphorylation, a mark of cytoplasmic retention and thus inactivity (Fig. 2e; and Supplementary Fig. 7). RNA-seq was next performed to assay expression of YAP transcriptional targets, using the YAP target gene list used to identify YAP as a mechanotransducer in 2D culture (Supplementary Table 1)[13]. In agreement with IF results, expression of YAP target genes did not trend with increased stiffness (Fig. 2f; and Supplementary Table 1) or col-1 density (Supplementary Fig. 8a). Notably, expression of canonical YAP target genes ANKRD1, CTGF, CYR61, and ITGB2 were not differentially regulated by enhanced stiffness (Fig. 2g) or col-1 density (Supplementary Fig. 8b). This is in contrast to the robust YAP activation and target gene expression demonstrated by the same MEC line in 2D culture (Supplementary Fig. 9)[27].

**Inducible CRISPR/Cas9 YAP knockout.** Given the surprising lack of YAP activation by increased 3D culture stiffness, we generated doxycycline (dox)-inducible CRISPR/Cas9 YAP knockout (ΔYAP) MCF10A cells to test the dispensability of YAP in mechanotransduction. As absence of YAP may impact cell growth in 2D culture prior to encapsulation, cells stably expressing dox-inducible Cas9 and sgRNA targeting YAP (MCF10A::Cas9/sgYAP) were first encapsulated in 3D culture without Cas9 induction. Following encapsulation, cells were treated with dox to induce Cas9 expression and sgRNA-directed editing. Dox treated MCF10A::Cas9/sg*YAP* cells showed depletion of YAP protein

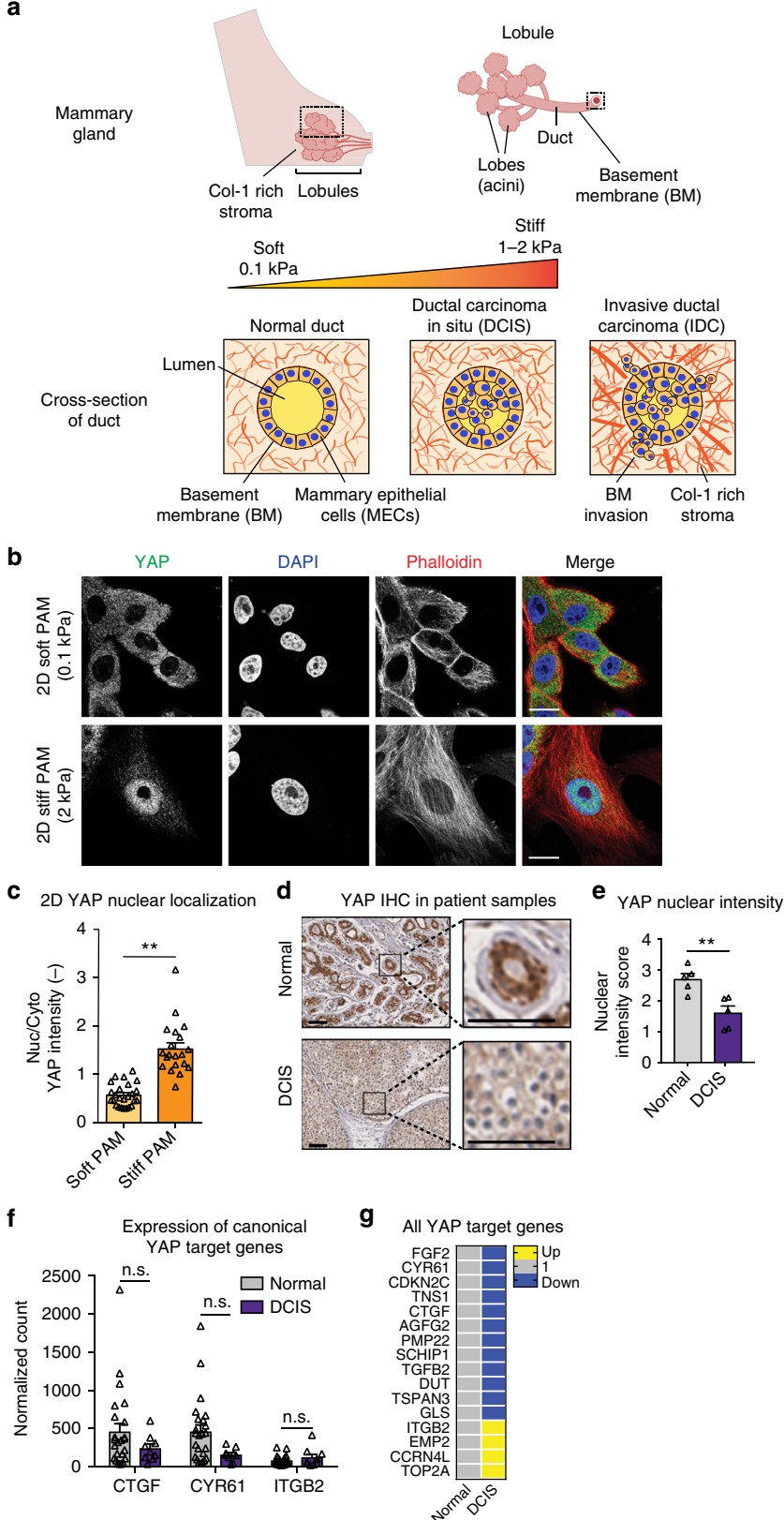

**Fig. 1** YAP is not activated during DCIS. **a** Schematic of ductal carcinoma progression. **b** MCF10A cells seeded on col-1-coated PAM gels. Bars: 10 μm.
**c** YAP quantification from 2D gels. **p < 0.0001, one-way ANOVA followed by Tukey post hoc comparison tests, symbols represent each cell, n = 9-14
cells per hydrogel. **d** YAP staining in primary tissues. Bars: 50 μm. **e** Quantification of YAP IHC intensity. **p < 0.01, unpaired t test, symbols represent each
patient sample, n = 5 normal and 5 DCIS patients. Expression of **f** canonical and **g** all YAP target genes in patient samples. n.s. not significant, one-way
ANOVA followed by Tukey post hoc comparison tests, symbols represent each patient sample, n = 24 normal and 9 DCIS patients. All bar charts display
mean ± SEM. All measurements were taken from distinct samples

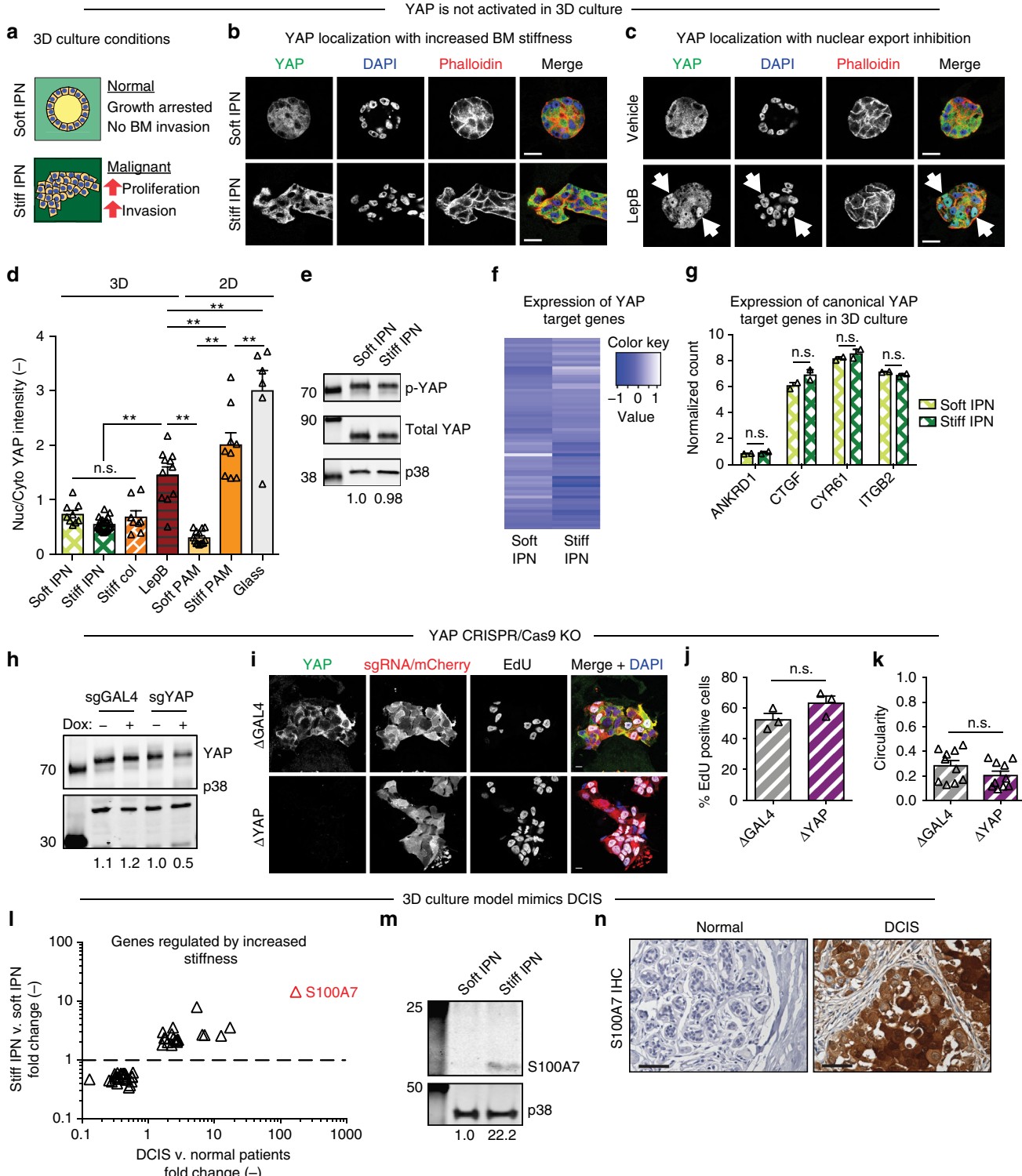

compared to untreated and MCF10A::Cas9/sg*GAL4* controls (Fig. 2h; and Supplementary Fig. 7). As Cas9 induction results in a mixed population of KO cells, only cells verified for ΔYAP by IF were assayed for mechanotransduction (Fig. 2i). Interestingly, *ΔYAP* cells did not reduce stiffness-induced invasion (Fig. 2j) or proliferation (Fig. 2k) compared to Δ*GAL4* controls. As YAP did not regulate mechanotransduction during breast cancer progression, we explored other transcriptional regulators whose target genes were identified by RNA-seq to be modulated by stiffness

(Supplementary Figure 10–12). Bioinformatics, small molecule inhibitor, inducible CRISPR/Cas9 KO, and overexpression experiments strongly implicate STAT3 and p300 as mechanotransducers during breast cancer (Supplementary Figs. 10 and 11). Taken together, our analyses of YAP and TAZ nuclear localization, YAP phosphorylation state, expression of YAP target genes, and inducible CRISPR/Cas9 knockout cells conclusively show that YAP does not mediate mechanotransduction in 3D culture.

**Fig. 2** Lack of YAP activation with enhanced 3D culture stiffness. **a** Effects of 3D culture stiffness. **b** MCF10A cells encapsulated in 3D hydrogels. **c** Encapsulated cells treated with Leptomycin B (LepB). Arrows indicate nuclei with YAP. Bars: 20 μm. **d** YAP quantification from 3D and 2D (control) culture conditions. **p < 0.05, one-way ANOVA followed by Tukey post hoc comparison tests, symbols represent each cell, $n = 6$–24 cells per hydrogel, bars represent mean ± SEM. Measurements were taken from distinct samples. **e** Western blot analysis of p-YAP (S127) from 3D culture. p38 was used as a loading control. Quantification of bands (p-YAP/total YAP/p38) below each lane. **f** RNA-seq of YAP target genes (as identified by Dupont et al., 2011) in 3D culture. **g** RNA-seq of canonical YAP target genes in 3D culture. n.s. not significant, unpaired two-sided t test, symbols represent each hydrogel, $n = 2$ hydrogels, bars represent mean ± SD. Measurements were taken from distinct samples. **h** Western blot analysis of dox-inducible MCF10A::Cas9/sgGAL4 or sgYAP cells. Quantification of bands (YAP/p38) below each lane. **i** CRISPR/Cas9 cells encapsulated with dox. Bars: 10 μm. **j** Proliferation of cells from **i**. **k** Invasiveness of cells from **i** as measured by cell cluster circularity. Only cells verified by IF for KO were assayed. **l** Set of genes regulated by enhanced stiffness in IPNs also upregulated in DCIS patient samples. Symbols represent individual genes. Most highly enriched gene (S100A7) highlighted in red. **m** Western blot analysis of cells from 3D culture for S100A7. Quantification of bands (S100A7/p38) indicated below each lane. **n** S100A7 staining in primary tissue. Bars: 50 μm

---

**Relevance of 3D culture model to breast cancer**. To assess the relevance of this 3D culture model to DCIS, we compared our RNA-seq data of cells encapsulated in soft or stiff IPNs (Fig. 2f) to 3SEQ data from normal and DCIS patient samples[28] (Fig. 2f, g). Importantly, a set of genes was identified that showed similar regulation in stiff IPNs as DCIS samples (Fig. 2l; and Supplementary Fig. 12 and Supplementary Table 2). Interestingly, RNA-seq of cells isolated from stiff col-1 containing gels show a distinct gene expression profile compared to BM stiffness, and captures key aspects of the gene-expression profile in IDC patient samples (Supplementary Fig. 12). Plotting fold change in vitro (i.e., stiff IPNs) against fold change in vivo (i.e., DCIS patient samples) revealed the most highly upregulated target from stiff IPNs, S100A7, as the most relevant stiffness-regulated gene in DCIS (Fig. 2l). S100A7 has been implicated in DCIS with roles in proliferation and apoptosis-resistance, and tumor-associated immune cell recruitment[29–31]. RNA-seq results were confirmed by WB analysis of S100A7 in cells harvested from soft and stiff IPNs (Fig. 2m; and Supplementary Fig. 7), IHC of S100A7 in breast cancer patient tissues (Fig. 2n), and qPCR of cells harvested from soft and stiff IPNs (Supplementary Fig. 13). Together, these results demonstrate that 3D culture of MECs in stiff IPNs is highly relevant to modeling DCIS, and provides a gene signature of stiffness-induced carcinoma progression.

**Cells in 3D culture and in vivo show decreased nuclear size**. To elucidate the mechanism underlying the confounding result that YAP is responsible for mechanotransduction in 2D, but not 3D culture nor primary tissue, we examined nuclear morphologies. This analysis was motivated by the recent finding that stiffness-induced YAP activation requires nuclear flattening and opening of nuclear pores[15,16]. Analysis of nuclear morphologies showed drastic differences in DCIS primary tissues and cells in 3D culture compared to 2D culture (Fig. 3a). Strikingly, nuclear area in cells from 2D culture show a tenfold increase in cross-sectional area compared to 3D culture and patient samples (Fig. 3b, Supplementary Fig. 6b). These changes in nuclear morphology also occur when cells are cultured on top of (2D) rather than encapsulated in (3D) the identical substrate: 20 kPa alginate–RGD hydrogels (Supplementary Fig. 6a–c).

A threefold increase in nuclear perimeter was also observed (Fig. 3c), in addition to a significant increase in solidity, a measure of the smooth nature of the perimeter (Supplementary Fig. 14a). During progression from normal to DCIS to IDC, patient samples showed interesting, but comparatively small, differences in nuclear morphology (Supplementary Fig. 14b–e).

Notably, YAP nuclear intensity scales with nuclear area, with nuclei from patient samples and 3D culture deviating from the size range observed for positive nuclear YAP intensity (Fig. 3d). Similarly, nuclear YAP intensity scales with nuclear perimeter (Fig. 3e). Positive nuclear YAP intensity also correlated with high

solidity, which is almost exclusively observed in nuclei from 2D culture (Supplementary Fig. 14f). However, some nuclei from soft 2D PAM reach the required size ranges but fail to show positive nuclear YAP intensity, suggesting that nuclear morphology is not the only factor required for YAP activation.

**MECs fail to form perinuclear stress fibers in 3D culture**. As nuclear morphologies and YAP activation in 2D culture has been linked to stress fiber contractility, we investigated the role of stress fibers in 3D culture. Recent studies showed mechanical coupling of stiff ECM to the nucleus through stress fibers, with fiber contractility causing nuclear flattening and subsequent nuclear pore stretching[15,16]. Further, the enrichment of perinuclear stress fibers was required for YAP nuclear translocation in 2D culture[16]. To examine if stress fibers contribute to the observed changes in nuclear morphology, we assayed stress fibers in cells cultured in 2D and 3D. The presence of robust perinuclear stress fibers was observed in cells cultured on stiff PAM gels, in which nuclear localization of YAP was observed, but not soft PAM gels nor stiff or soft IPNs, in which nuclear localization of YAP was not observed (Fig. 3f, g). This is in agreement with previous reports that cells cultured in 3D substrates fail to form robust stress fibers[32,33], and instead adopt a predominantly cortical F-actin architecture (Fig. 3g). This suggests a model where the presence of perinuclear stress fibers, coupled with distinct nuclear morphologies, is the basis of differences between YAP 2D and 3D activation (Fig. 3h).

In this study, we examined the role of YAP in mediating mechanotransduction during ductal carcinoma progression using patient samples and 3D culture models. Cancer has historically been thought of as a genetic disease, with tumors arising from genetic mutations in DNA. However, it has been increasingly recognized that the microenvironment plays a key role in regulating cancer progression. Our study provides compelling evidence that 2D YAP mechanotransduction studies do not recapitulate the conditions seen in clinical samples, and suggests a critical need for the use of 3D culture models in studying breast cancer. Finally, our findings reveal new therapeutic targets, including STAT3 and p300, for preventing breast cancer invasion.

## Methods

**Cell culture and cell lines**. MCF10A cells obtained from the ATCC (cat. #CRL-10317; ATCC) were cultured in DMEM/F12 50/50 medium (cat. #11330057; Thermo Fisher Scientific) supplemented with 5% horse serum (cat. #16050122; Thermo Fisher Scientific), 20 ng/ml EGF (cat. #AF-100-15; Peprotech, Inc.), 0.5 μg/ml hydrocortisone (cat. #H0888-1G; Sigma), 100 ng/ml cholera toxin (cat. #C8052-1MG; Sigma), 10 μg/ml insulin (cat. #91077C-250MG; Sigma), and 100 U/ml Pen/Strep (cat. #15140; Thermo Fisher Scientific) as previously described[28]. MCF10AT and MCF10CA1a cells were a gift from Lalage Wakefield (NIH) and were cultured in complete medium for experimental consistency.

For inducible MCF10A::Cas9 cell line, lentivirus was produced harboring Edit-R Inducible Lentiviral hEF1α-Blast-Cas9 Nuclease Plasmid DNA (cat. #CAS11229; Dharmacon) (see Cloning and lentiviral generation below). Following infection,

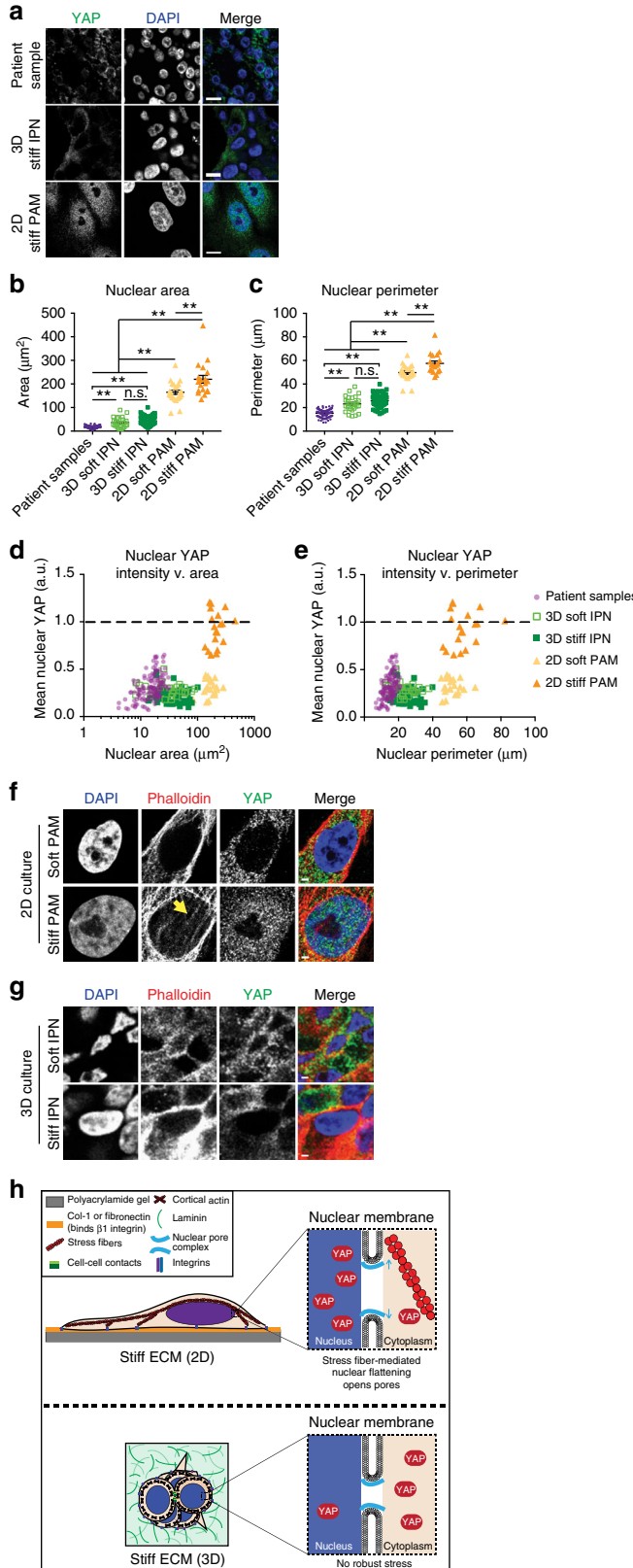

Fig. 3 Nuclear morphologies are distinct between 2D and 3D culture and in vivo. **a** Images of nuclear morphologies and YAP. Bars: 10 μm. **b** Areas and **c** perimeters of nuclei. **$p < 0.005$, one-way ANOVA followed by Tukey post hoc comparison tests, symbols represent each cell, $n = 19-132$ nuclei per condition, lines display mean ± SEM. Measurements were taken from distinct samples. Patient samples from five DCIS patients. YAP intensity with nuclear **d** area and **e** perimeter. Values normalized by positive controls within each sample. Dotted line represents positive nuclear YAP. Purple dots: patient samples; light green squares: 3D soft IPN; filled dark green squares: 3D stiff IPN; filled light orange triangles: 2D soft PAM; filled orange triangles: 2D stiff PAM. F-actin staining in **f** 2D or **g** 3D culture. Arrow indicates perinuclear stress fibers. Bars: 2 μm. **h** Model of stiffness-induced YAP localization in 2D vs. 3D

**Reagents**. EdU incorporation assay (cat. #C10337; Thermo Fisher Scientific) was performed according to manufacturer's instructions with a 24 h incubation of 10 μM EdU.

For inhibitor studies, MCF10A cells encapsulated in hydrogels were incubated in Genentech p300 inhibitor GNE-049 at a final concentration of 0.5 μM (Genentech, Inc.; MTA OR# 216339) or STAT3 peptide inhibitor PY*LKTK at a final concentration of 500 μM (cat. #ab142104; Abcam). p300 inhibitor C646 (cat. #SML0002; Sigma) was also used at the indicated concentrations.

CRISPR/Cas9 MCF10A cell lines were generated by first producing a doxycycline-inducible Cas9/blast MCF10A cell line. Cas9/blast transfer vector (cat. #CAS11229; Dharmacon) containing virus was produced and used to infect WT MCF10A cells. Stably expressing cells were selected using 5 μg/ml blast. MCF10A::Cas9/blast cells were infected with lentivirus harboring sgRNA against GAL4/mCherry/puro, YAP/mCherry/puro, or STAT3/mCherry/puro. Doubly stably expressing cells were selected using 5 μg/ml blast with 1 μg/ml puro. CRISPR/Cas9 editing was induced by adding 2 μg/ml dox (cat. #AAJ6042206; Alfa Aesar) and knockout verified by WB and IF.

**Cloning and lentiviral generation**. All plasmids in this study have been banked in Addgene using the following ID numbers:
pLenti-GAL4 (121514), pLenti-sgYAP-2 (121423), pLenti-sgYAP-10 (121424), pLenti-sgSTAT3-1 (121425), pLenti-EGFP (121426), and pLenti-STAT3-linker-EGFP (121427).

sgRNAs were expressed using a lentiviral mouse U6 (mU6) promoter-driven expression vector that coexpessed Puro-T2A-mCherry from a CMV promoter. sgRNA sequences were generated by PCR and introduced by InFusion cloning into the sgRNA expression vector digested with BstXI and XhoI. sgSTAT3-1 sequence: GTCAGGATAGAGATAGACCAG. For YAP, 2 sgRNA sequences were used and pooled during lentiviral production. sgYAP-2 sequence: GAGATGACTTCCTGAACAGTG; sgYAP-10 sequence: GTGCTGTCCCAGATGAACGTC.

To assemble pLenti-STAT3-linker-EGFP for overexpression, STAT3 was amplified from pLEGFP-WT-STAT3, with the forward primer containing the linker sequence, and inserted using Infusion Cloning into MluI and EcoRI digested pLenti-Origene-Nrf21. To assemble pLenti-EGFP control, EGFP was amplified from pLenti-Origene-Nrf21 and inserted into XhoI and EcoRI digested pLenti-Origene-Nrf21.

For lentiviral generation, HEK293T cells were seeded at $1 \times 10^7$ cells/10 cm dish. The next day 70–90% confluent cells were transfected. For each dish, 9 μg of lentiviral transfer vector, 8 μg of packaging vector pCMV-dR8.91, and 1 ug of packaging vector pMD2-G were transfected using Lipofectamine 3000 Transfection Reagent (cat. #L3000008; ThermoFisherScientific) Opti-MEM Reduced Serum Medium (cat. #31985062; Gibco) according to the manufacturer's instructions. Medium was replaced with complete medium 4 h following transfection. Forty-eight hour following transfection, lentivirus-containing supernatant was harvested and filtered through a 0.22 μm Steriflip (Millipore). Filtered supernatant was concentrated using Lentivirus Precipitation Solution (cat. #VC100; AlStem) according to the manufacturer's instructions. Following concentration, lentiviral pellets were resuspended in 1/100 of original volume using cold DMEM/F12 and stored at −80 °C. For MCF10A transduction, concentrated lentivirus was added to complete medium containing 8 μg/ml polybrene (cat. #SC134220; Santa Cruz Biotech) at a volume of 1:100.

**Hydrogel formation**. Matrigel (cat. #354230; Corning) was purchased for use as rBM matrix and used at a final concentration of 4.4 mg/ml for all experiments. Collagen-1, derived from rat tail, (cat. #354236; Corning) was lyophilized and reconstituted in 20 mM acetic acid. Immediately before cell encapsulation, reconstituted col was supplemented with 10× PBS, neutralized with 0.1 M NaOH, and pH adjusted with 0.1 N HCl. rBM and col were mixed with cells and DMEM/F12 to the reach the indicated final concentrations. MCF10A cells were trypsinized, strained through a 40 μm cell strainer to enrich for single cells, counted on a Vi-CELL (Beckman Coulter Life Sciences), and seeded at a final concentration of

Cas9 cells were maintained in MCF10A growth medium as above, supplemented with 5 μg/ml blasticidin (cat. #R21001; Thermo Fisher Scientific). Following a second round of infection with indicated sgRNAs (see Cloning and lentiviral generation below)., MCF10A::Cas9/sgRNA cell lines were maintained in medium supplemented with 5 μg/ml blasticidin and 1 μg/ml puromycin (cat. #A1113803; Thermo Fisher Scientific).

$1 \times 10^5$ cells/ml hydrogel. Hydrogel-cell mixtures were quickly deposited into wells of a 24-well plate precoated with 50 µl gelled rBM. Hydrogels containing cells were placed in a 37 °C incubator with $CO_2$ to gel for 30 min before a transwell insert (Millipore) was placed on top to prevent floating and 1.5 ml complete medium added.

IPNs were formed as previously described[10]. LF20/40 alginate (FMC Biopolymer) was solubilized, dialyzed, charcoal filtered, sterile filtered, lyophilized, and reconstituted to 2.5% w/v in DMEM/F12. Alginate was mixed with rBM, cells, and DMEM/F12 and loaded into a 1 ml Luer lock syringe (Cole-Parmer), on ice. For crosslinking, a second 1 ml syringe was loaded with 125 mM $CaSO_4$ or DMEM/F12, on ice. Syringes were connected with a female–female Luer lock coupler (ValuePlastics), rapidly mixed with four to six pumps of the syringes handles back and forth, and quickly deposited into precoated wells, as above. IPNs containing cells were allowed to gel before adding transwell filters and medium, as above.

For 20 kPa alginate–RGD hydrogels, alginate–RGD hydrogels were prepared as described previously[34]. LF20/40 alginate (FMC Biopolymer) was dialyzed, filtered and lyophilized, and then was coupled to RGD oligopeptide GGGGRGDSP (Peptide 2.0) using carbodiimide chemistry[35]. The final density of RGD in the alginate hydrogel was matched as 150 mM RGD in a 2% wt/vol alginate gel. The modified alginate was dialyzed, charcoal filtered, sterile filtered and lyophilized again. Alginate–RGD was reconstituted to 2.5% w/v in DMEM/F12 and mixed with MCF10A cells. The cell–alginate solution was then mixed with DMEM/F12 containing 24.4 mM $CaSO_4$, and then deposited between two glass plates spaced 2 mm apart. The cell–alginate mixture was allowed to gel for 45 min, and then disks of hydrogel were punched out and immersed in complete growth medium.

For 2D PAM gels, the surface of coverslips was functionalized accoring to a previous method[36]. Coverslips were cleaned with ethanol, immersed in 0.5% (3-aminopropyl)trimethoxysilane (in $dH_2O$) at room temperature for 30 min and washed with $dH_2O$. Coverslips were then immersed in 0.5% glutaraldehyde in $dH_2O$ at room temperature for 30 min and dried. A prepolymer solution was prepared containing acrylamide, N,N′-methylene-bis-acrylamide, 1/100 volume of 10% ammonium persulfate, and 1/1000 volume of N,N,N′,N′-tetramethylethylenediamine (TEMED). The final concentration of acrylamide and bis-acrylamide was varied to control substrate stiffness[37]. For 0.1 kPa hydrogels, 3%/0.02% was used. For 1 kPa hydrogels, 3%/0.1% were used. For 2 kPa hydrogels, 4%/0.1% were used. Prepolymer solutions were deposited on a Sigmacote-treated hydrophobic glass plate, and functionalized coverslips placed on top of the prepolymer solution. Polyacrylamide solutions were allowed to polymerize for 30 min between the hydrophobic glass plate and the functionalized coverslip. When polymerization was completed, PAM gels were carefully separated from the glass plate.

To enable cell adhesion to the PAM gel, col-1 and rBM were conjugated to the gel surface using sulfosuccinimidyl 6-(4′-azido-2′-nitrophenylamino)hexanoate (sulfo-SANPAH) as a protein-substrate linker. PAM gels were incubated in 1 mg/ml sulfo-SANPAH in 50 mM HEPES pH 8.5, activated with UV light (wavelength 365 nm, intensity 4 mW/cm$^2$) for 20 min, washed in HEPES, and then incubated in 100 µg/ml col-1 and rBM in HEPES overnight at room temperature. The protein cross-linked PAM gels were washed with PBS before use.

**Mechanical testing.** Stiffness measurements of 3D culture rBM, col, and IPN hydrogels were performed using an AR-G2 stress-controlled rheometer with 25-mm top- and bottom-plate stainless steel geometries (TA Instruments). Hydrogel solutions without cells were mixed and immediately deposited onto the bottom plate of the rheometer and the top plate lowered such that the gel formed a uniform disk between the two plates. Exposed hydrogel surfaces were coated with mineral oil (Sigma) and covered with a hydration chamber to prevent sample dehydration. The storage modulus was monitored at 37 °C with 1% strain at a frequency of 1 Hz and measurements taken once the storage modulus reached an equilibrium value. The storage and loss moduli were then used to calculate the Young's modulus ($E$). Young's moduli (i.e., elastic moduli) were calculated using the equation $E = 2 G^* \times (1 + v)$, where $v$ is Poisson's ratio, assumed to be 0.5, and $G^*$ is the bulk modulus calculated using the equation $G^* = (G'^2 + G''^2)^{1/2}$ where $G'^2$ is the storage and $G''^2$ is the loss modulus.

To measure substrate stiffness of 2D PAM gels, unconfined compression tests were performed using an Instron MicroTester 5848. PAM gels were compressed at a rate of 1 mm/min. The tangent elastic modulus of the measured stress-strain curves was calculated between 5 and 15% strain[4,7]. Stiffness of 3D culture alginate hydrogels was measured using unconfined compression tests according to a previously published method[34]. Alginate disks (15 mm in diameter, 2 mm thick) were submerged in DMEM for 1 day to fully equilibrate. The gel disks were compressed to 15% at a rate of 1 mm min$^{-1}$ and the slope of the stress–strain curve from 5% to 10% strain was used to obtain the stiffness of alginate hydrogel.

**Antibodies.** Mouse anti-YAP (cat. #sc-101199; Santa Cruz Biotech) was used at 1:200 (IF) and 1:500 (WB), rabbit anti-phospho-YAP (cat. #13008; Cell Signaling Technology) was used at 1:500 for WB. Mouse anti-S100A7 (cat. #sc-377084; Santa Cruz Biotech) was used at 1:500 (WB). Mouse anti-S100A7 (cat. #HPA006997; Millipore-Sigma) was used at 1:100 for IHC. Rabbit anti-phospho p300 (cat. #ab135554; Abcam) was used at 1:500 for WB, mouse anti-p300 (cat. #sc-32244; Santa Cruz Biotech) was used at 1:500 for WB. Rabbit anti-phospho STAT3 (cat.

#ab76315; Abcam) was used at 1:500 for WB, mouse anti-STAT3 (cat. #sc-8019; Santa Cruz Biotech) was used at 1:500 for WB. Rabbit anti-p38 (cat. #sc-535; Santa Cruz Biotechnology) was used at 1:2000 for WB. Mouse anti-β1 integrin (cat. #ab24693, Abcam) was used at 1:500 for IHC. Rabbit anti-phospho FAK (cat. #31H5L17; Thermo Fisher Scientific) was used at 1:100 for IHC. Rabbit anti-α-actinin (cat. #3134; Cell Signaling Technology) was used at 1:500 for WB. Rabbit anti-β-actin (cat. #8457; Cell Signaling Technology) was used at 1:1000 for WB. Phalloidin-Alexa555 (A34055; Thermo Fisher Scientific) was used at 1:100 and DAPI (cat. #D9542; Sigma-Aldrich) was used at 5 µg/ml for IF.

For IF, Alexa 488-, 555-, or 647-conjugated secondary antibodies (Thermo Fisher Scientific) were used at 1:500. For WB, IRDye 680 or 800-conjugated secondary antibodies (LI-COR Biotechnology) were used at 1:10,000.

**WB and immunoprecipitation.** Uncropped WBs shown in Supplementary Fig. 7. MCF10A cells encapsulated for 7 days were harvested from IPNs by incubation in cold PBS containing 50 mM EDTA (Sigma) for 5 min while pipetting to break up gels. Cells were centrifuged at 500$g$ for 10 min. The supernatant was removed and the cells with remaining matrix material were treated with 0.25% trypsin (Gibco) for 5 min and centrifuged for 5 min at 500$g$. Cell pellets were washed with 20% serum-containing resuspension buffer to neutralize trypsin and washed twice with PBS. For sodium dodecyl sulfate polyacrylamide gel electrophoresis of whole-cell lysates, MCF10A cells were lysed in Pierce RIPA buffer (cat. #89900; Thermo Fisher Scientific) supplemented with Protease Inhibitor Cocktail Tablets (cat. #11836170001; Roche) and PhosSTOP Phosphatase Inhibitor Cocktail Tablets (cat. #04906845001; Roche) according to the manufacturer's instructions. Protein concentration was determined using the Pierce BCA Protein Assay Kit (cat. #23227; Thermo Fisher Scientific). Laemmli sample buffer (cat. #1610747; Bio-Rad) was added to lysates and samples boiled for 10 min before loading 25 µg protein in each lane of a 4–15%, 15-well, gradient gel (cat. # 4561086; Bio-Rad). Proteins were transferred to nitrocellulose at 100 V for 105 min, blocked with 5% milk in TBS-T (137 mM NaCl, 2.7 mM KCl, 19 mM Tris base, 0.1% Tween, pH 7.4), incubated in primary antibody overnight, IRDye 680- or 800-conjugated secondary antibodies (Li-COR Biotechnology) for 1 h, and visualized with the Li-COR Odyssey imaging system (Li-COR Biotechnology). Quantitative analysis of western blots was performed using the Li-COR Odyssey software (LI-COR Biotechnology).

**Immunofluorescence.** Cells encapsulated in hydrogels for seven days were fixed for 30 min in 4% paraformaldehyde in DMEM/F12. Gels containing cells were washed with PBS and incubated in 30% sucrose in PBS with calcium and magnesium overnight followed by incubation in 50/50 30% sucrose/OCT for 6 h. Gels were embedded in OCT and frozen prior to cutting 40 µm sections using a Microm HM 550 Cryostat. Sections were blocked in PBS-BT+: PBS pH 7.4 (Gibco) supplemented with 1% bovine serum albumin (Sigma-Aldrich), 0.1% Triton X-100 (Sigma-Aldrich), 0.3 M glycine (Sigma-Aldrich), 10% goat serum (Gibco), and 0.05% sodium azide (Sigma-Aldrich). Sections were incubated in primary antibodies diluted in blocking solution as indicated in "antibodies" section for 1 h, and then Alexa 488-, 555- or 647-conjugated secondary antibodies (Thermo Fisher Scientific) diluted 1:500 in blocking solution for 30 min. Sections of gels containing cells were imaged using a Leica TCS SP8 confocal laser scanning microscope (Leica Microsystems, Inc.) with an HC PL APO 63× (1.40 NA Oil CS2) objective. Images were collected from HyD and PMT detectors using LasX software and processed using Photoshop (Adobe Systems).

For morphology analyses, ImageJ was used to trace cell clusters and circularity measured using the Measurements function. Circularity, $C$, was calculated as, $C = 4\pi(A/p^2)$, where $A$ is the area and $P$ is the perimeter. A perfect circle would have a circularity of 1. Solidity was calculated as area enclosed by outer contour of object divided by area enclosed by convex hull of outer contour.

Cell Profiler was used to quantify YAP nuclear/cytoplasmic intensity in IF images. ImageJ was used to trace cell nuclei using DAPI images using the following macro (pixel/µm of image was first determined and replaced in "Set Scale" distance; found pixel/µm of image by drawing line over scale bar embedded in image and using the function Analyze - > Set Scale). Doublets or cell debris were then manually excluded. Nuclear traces were then overlaid on YAP images to measure mean nuclear YAP intensity using the following macro.

Macro to trace cell nuclei:
```
run("Set Scale…", "distance = [3.45] known = 1 pixel = 1 unit = µm");
run("Gaussian Blur…", "sigma = 2");
run("Subtract Background…", "rolling = 50");
setAutoThreshold();
//run("Threshold…");
setAutoThreshold();
setThreshold(55, 255);
run("Convert to Mask");
run("Fill Holes");
run("Watershed");
run("Find Edges");
run("Analyze Particles…", "size = 100-Infinity pixel circularity =
0.00–1.00 show = Nothing exclude add");
close();
```
Macro to measure nuclear YAP intensity:

```
run("Set Measurements…", "area mean center perimeter bounding shape
integrated skewness redirect = None decimal = 3");
run("Set Scale…", "distance = [3.45] known = 1 pixel = 1 unit = μm");
setOption("Show All",true);
roiManager("Measure");
saveAs("Measurements", "/Users/Joanna/Desktop/Results.xls");
```

**Tissue immunohistochemistry.** IHC staining was performed on paraffin-embedded tissue microarray (TMA) sections (Stanford TA419, 445, 424). Anti-S100A7 polyclonal antibody at 1:100 (Millipore-Sigma St. Louis, MO, catalog # HPA006997) was used as primary antibody for IHC staining. Antibody was diluted in PBS. TMA sections measuring 4 μm were deparaffinized in 3 changes of xylene for 10 mins each and hydrated in gradient series of ethyl alcohol. Following target retrieval in 10 mM citrate pH6 (Dako/Agilent, Carpinteria, CA, USA, catalog #S2369) to retrieve antigenic sites at 116 °C for 3 min. Staining was then performed using the VectaStain ABC anti-rabbit kit (Vector Laboratories Burlingame CA, USA, catalog #PK6101). Diaminobenzidine (DAB) (DAKO/Agilent, Carpinteria, CA, USA, catalog #K3468) was used at room temperature for 10 min for color development. The IHC Profiler macro for ImageJ was used to quantify YAP and S100A7 intensity in IHC samples[38].

**RNA extraction and next generation sequencing.** Gels containing MCF10A cells were frozen in liquid nitrogen, ground, and treated with ice-cold PBS supplemented with 50 mM EDTA to break up IPNs. RNA was harvested using a combination of TRIZOL reagent and GenCatch Total RNA Extraction Kit (Epoch). RNA-seq experiments were performed in biological replicate and cDNA libraries constructed using the TruSeq RNA Library Prep Kit v2. Libraries were sequenced on a single lane of the Illumina Hiseq 2500 platform with 50 bp paired end reads. Following quality assessment via FastQC, adapter and quality trimming was executed with Trim Galore. Reads were subsequently aligned to the *hg19* genome assembly via Bowtie2 with ~97% concordant alignment rate in all samples. After filtering for unmapped, low quality, and multimapped reads, mapped reads were summarized to gene features by HTSeq. Sequencing depth ranged from 30 to 42 million postfiltered reads. We used DESeq2 to evaluate significant cases of differential expression between a given pairwise comparison. Before adjusting p-values for multiple testing, DESeq2 implements independent filtering using mean expressions of each gene as a filter. Adjusting via the Benjamini & Hochberg method, differentially expressed genes with FDR < 0.05 were called significant. Prior to clustering, we used DESeq2's implementation of regularized logarithm transformation (rlog) to stabilize the variance of genes across samples. Mean expression values were used as input to hierarchical clustering of the differentially expressed genes between soft v. stiff IPN. Gene ontology and TF association analysis using ChIP-seq data from ENCODE was implemented via EnrichR[39]. Adjusted p values, which take into account differing sizes of data sets, are reported.

**Statistical analysis.** Multiple comparisons were conducted with one-way ANOVA followed by Tukey post hoc comparison and pairwise comparisons performed using Student's *t* tests. Bars represent mean ± SEM and symbols represent each experiment (for EdU assays), cell cluster (for invasion assays), patient (for 3SEQ expression), and nucleus (for nuclear morphology assays). For graphs of RNA-seq normalized counts from 3D culture, bars represent mean ± SD and symbols represent each RNA-seq experiment using two independent trials. Values with $p < 0.05$ were considered statistically significant, indicated by **.

**Reporting summary.** Further information on research design is available in the Nature Research Reporting Summary linked to this article.

## Data availability

RNA-seq data are stored in GEO with the accession code GSE102506. 3SEQ breast cancer progression data are available as described in the original manuscript[28]. Additional normal mammary and breast cancer datasets generated by the Genotype-Tissue Expression (GTEx) project and The Cancer Genome Atlas (TCGA) project, respectively, are available on the Human Protein Atlas[40,41]. Additional datasets generated during and/or analyzed during the current study are available from the corresponding author on reasonable request.

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

## Acknowledgements

We gratefully acknowledge Katrina Wisdom (Stanford University) and Ryan Stowers (Stanford University) for assistance with mechanical testing, materials, and helpful discussions. We acknowledge Robert Blake (Genentech) for helpful advice and assistance with p300 inhibitors. We thank Lalage Wakefield and Binwu Tang (NIH) for assistance with MCF10AT and MCF10CA1a cell lines. We thank Sasha Denisin (Stanford University) and Bayana Science for careful reading of the manuscript. This work was supported by an American Cancer Society Grant (RSG-16-028-01) and a National Institutes of Health grant (R37 CA214136) to OC.

## Author contributions

Conceived and designed the experiments: J.Y.L. and O.C. Performed the experiments: J.Y.L., J.K.C., A.A.D., H.L., S.V., and J.C. Analyzed the data: J.Y.L., J.K.C., and O.C. Contributed reagents/materials/analysis tools: J.Y.L., J.K.C., A.A.D., H.L., S.N., R.B.W., and O.C. Supervised and administered the project: L.S.Q., R.B.W., and O.C. Acquired funding: O.C. Wrote the paper: J.Y.L. and O.C.

## Additional information

**Competing interests:** L.S.Q. received sponsored research support from Tencent America Inc. and is a co-founder and stock shareholder of Refuge Biotechnologies. The remaining authors declare no competing interests.

