## [Peer Review File · Nature Communications]

REVIEWERS' COMMENTS:

Reviewer #1 (Remarks to the Author):

The revised manuscript by Lee et al. have made major efforts in focusing the manuscript on the key message that YAP is dispensable for mechanotransduction in 3D mammary organoids, which is very different from what have been reported in 2D cultures. The revision is much more focused and included interesting new data to support this conclusion. The most interesting new data is the suggestion that nuclear morphologies are very different between cells in 2D vs. 3D culture, which could cause the different YAP responses. Although these new data remain correlative and do not provide definitive molecular mechanisms to explain these interesting observations, these data should be published given the somewhat simplistic view of YAP in mechanotransduction in current literatures. Therefore, I support the publication of this study.

However, one remaining point brought up in previous review should still be addressed. One possibility is that the 2D vs. 3D culture conditions results in different nuclear morphologies. However, the current comparison between 2D vs. 3D uses two different gel systems, IPN in 3D vs. PAM in 2D. The collagen system also changes the collagen concentration, thus not being able to differentiate biochemical vs. mechanical changes. This is why previous review suggested to perform the analysis on 3D culture on PAM gels to directly compare the same matrix material in 2D vs. 3D culture. Although full embedding with PAM gel is not feasible, it has been performed in many studies using the overlay 3D culture with Matrigel-embedded organoids on top of the PAM gel. Along the same line, cells could be cultured on top of 2D 20kPa alginate-RGD gel vs. 3D culture presented in Fig. S6B to compare. These are very straightforward and feasible experiments, These comparisons are essential to ensure that the observed differential YAP responses are indeed physiological and not due to the different types of synthetic hydrogels used for the study.

RESPONSE TO REVIEWER

Reviewer #1 (Remarks to the Author):

The revised manuscript by Lee et al. have made major efforts in focusing the manuscript on the key message that YAP is dispensable for mechanotransduction in 3D mammary organoids, which is very different from what have been reported in 2D cultures. The revision is much more focused and included interesting new data to support this conclusion. The most interesting new data is the suggestion that nuclear morphologies are very different between cells in 2D vs. 3D culture, which could cause the different YAP responses. Although these new data remain correlative and do not provide definitive molecular mechanisms to explain these interesting observations, these data should be published given the somewhat simplistic view of YAP in mechanotransduction in current literatures. Therefore, I support the publication of this study.

However, one remaining point brought up in previous review should still be addressed. One possibility is that the 2D vs. 3D culture conditions results in different nuclear morphologies. However, the current comparison between 2D vs. 3D uses two different gel systems, IPN in 3D vs. PAM in 2D. The collagen system also changes the collagen concentration, thus not being able to differentiate biochemical vs. mechanical changes. This is why previous review suggested to perform the analysis on 3D culture on PAM gels to directly compare the same matrix material in 2D vs. 3D culture. Although full embedding with PAM gel is not feasible, it has been performed in many studies using the overlay 3D culture with Matrigel-embedded organoids on top of the PAM gel. Along the same line, cells could be cultured on top of 2D 20kPa alginate-RGD gel vs. 3D culture presented in Fig. S6B to compare. These are very straightforward and feasible experiments, These comparisons are essential to ensure that the observed differential YAP responses are indeed physiological and not due to the different types of synthetic hydrogels used for the study.

We are grateful to the reviewer for the supporting publication of our manuscript and for the important suggestion. We have now performed the experiment requested by the reviewer. MCF10A cells were cultured on top of 20 kPa alginate-RGD hydrogels (2D culture), and nuclear morphologies and YAP localization was directly compared to MCF10A cells cultured within alginate-RGD hydrogels of the same formulation (3D culture). We find that cells cultured in 2D show YAP nuclear localization and larger nuclear cross-sectional areas, confirming the main findings from the manuscript. We have included these new data in Supplementary Figure 6b, c and in the main text. The relevant figure panels also included in the following page.

Supplementary Figure 6. YAP staining with 20 kPa stiffness, MCF10AT and MCF10CA1a cells, and transplantations. (A) Unconfined compression of hydrogels compressed to 15% at a rate of 1 mm min^{-1} to obtain the stiffness of alginate hydrogels. Bars represent mean of four gels \pm SEM, symbols represent E of each gel. (B) Nuclear areas of MCF10A cells cultured under 2D and 3D conditions with alginate-RGD. Dot plot lines display mean \pm SEM. **, $p < 0.0001$; unpaired t-test. (C) MCF10A cells encapsulated in 3D for 3 days or in 2D for 1 day using 20 kPa alginate-RGD hydrogels. Scale bars: $10 \mu\text{m}$.